# Tyrosol Derivatives, Bearing 3,5-Disubstituted Isoxazole and 1,4-Disubstituted Triazole, as Potential Antileukemia Agents by Promoting Apoptosis

**DOI:** 10.3390/molecules27165086

**Published:** 2022-08-10

**Authors:** Zaineb Abdelkafi-Koubaa, Imen Aissa, Hichem Ben Jannet, Najet Srairi-Abid, Naziha Marrakchi, Samia Menif

**Affiliations:** 1Laboratory of Biomolecules, Venoms and Theranostic Applications, LR20IPT01, Pasteur Institute of Tunis, University of Tunis El Manar, Tunis 1002, Tunisia; 2Team Medicinal Chemistry and Natural Products (LR11ES39), Laboratory of Heterocyclic, Chemistry, Natural Products and Reactivity, Department of Chemistry, Faculty of Science of Monastir, University of Monastir, Monastir 5019, Tunisia; 3Faculty of Medicine of Tunis, University of Tunis El Manar, Tunis 1068, Tunisia; 4Molecular and Cellular Hematology Laboratory, LR16IPT07, Pasteur Institute of Tunis, University of Tunis El Manar, Tunis 1002, Tunisia

**Keywords:** chronic myeloid leukemia, tyrosol, 3,5-disubstituted isoxazole, 1,4-disubstituted triazole, anti-proliferative activity, oxidative stress, apoptosis

## Abstract

In the present study, we assess tyrosol derivatives bearing 3,5-disubstituted isoxazoles and 1,4-disubstituted triazoles for their ability to inhibit the proliferation of K562 cells derived from leukemia as well as primary chronic myeloid leukemia (CML) cells obtained from the peripheral blood of 15 CML patients including 10 patients with untreated chronic phase and 5 patients with resistance against imatinib or multiple TKI. Our results showed that most derivatives displayed significant anti-proliferative activity against K562 cells in a dose-dependent manner. Among them, compounds **3d** and **4a** exhibited greater potent anticancer activity with respective IC_50_ values of 16 and 18 µg/mL (45 µM and 61 µM). Interestingly, compound **3d** inhibited CML cell proliferation not only in newly diagnosed but also in imatinib-resistant patients. We demonstrated that the anti-proliferative effect of this compound is mediated by a pro-apoptotic activity by promoting oxidative stress and modulating the activity of the Akt, p38 MAPK and Erk 1/2 pathways. In conclusion, our data highlight the potential of this class of derivative as a novel promising therapeutic agent for CML therapy.

## 1. Introduction

Chronic myeloid leukemia (CML) is a clonal myeloproliferative neoplasm that originates from hematopoietic stem cells (HSCs). It affects annually 1–2 per 10^5^ adults worldwide [1]. CML is driven by the bcr–abl fusion gene generated from the t(9;22)(q34;q11) translocation, resulting in a derivative, 22q-, traditionally known as the Philadelphia chromosome [2].

CML is one of the few malignant diseases triggered by a single oncogene, bcr–abl, encoding an abnormal protein with constitutive tyrosine kinase activity, responsible for proliferative and anti-apoptotic signals. This is the reason for the relative efficacy of molecular-targeted therapy.

Indeed, tyrosine kinase inhibitors (TKIs) were discovered in 1996 and led to a paradigm shift in the management of CML in which the universally fatal disease became merely a chronic illness, if managed appropriately [3]. However, despite their efficacy in the treatment of CML, long-term exposure to TKIs is generally associated with chronic side effects, the emergence of resistant phenotypes and a financial burden for health care systems [4].

This has, therefore, raised the need for the development of new compounds that can improve the effect of tyrosine kinase inhibitors and overcome drug resistance.

In this context, plant-derived natural products play essential roles in the development of anticancer agents and continue to constitute new leads in clinical trials [5]. Indeed, the search for plant-derived compounds with anticancer activity has received much attention due to the fact of their structural diversity, favorable pharmacokinetic properties and synthetic availability [6]. Over the last decades, heterocyclic compounds and their derivatives have attracted considerable attention in agrochemistry, medicinal chemistry and other emerging areas [7]. The development of novel and efficient synthesis protocols for attractive heterocyclic scaffolds has become an important goal in modern organic synthesis [8,9]. Triazole/isoxazole derivatives are a class of *N*-heterocyclic compounds with a broad range of biological and pharmacological activities including antibacterial and anticancer properties [10,11,12,13,14]. Particularly, some isoxazole and triazole derivatives have exhibited potent cytotoxic effects against various leukemia cell lines [15,16,17,18]. Through their remarkable biological activity, numerous isoxazoles and triazoles have already been used in clinics or are under clinical evaluation for cancer treatment, revealing their potential as putative anticancer drugs [19,20].

In a previous study, we reported on the synthesis of two series of 3,5-disubstituted isoxazole (**3a**–**e**) and 1,4-disubstituted triazole (**4a**–**e**) derivatives. Some derivatives inhibited significantly the proliferation of the U87 glioblastoma cell line with a pro-apoptotic activity [21], promoting a possible path for their further exploitation.

Based on these interesting results, we were encouraged to investigate their antileukemia effect, since our Laboratory of Molecular and Cellular Hematology at the Pasteur Institute of Tunis is a national referral center for all suspected or confirmed Tunisian leukemia patients and receives specimens from patients newly diagnosed or being treated to conduct hematologic, cytogenetic and molecular examinations.

Thus, all compounds were first evaluated for their effect on the K562 cell line, which has been considered as a cellular model of CML for drug screening. Then, we investigated the potential activity of selected compounds on primary CML patients’ cells and explored the mechanism of action of the most potent anti-proliferative compound. The results of the present study could provide insight into tyrosol derivatives in the development of a novel therapeutic target in the management of CML.

## 2. Material and Methods

### 2.1. Chemical Reagents

Two series of tyrosol derivatives bearing 3,5-disubstituted isoxazole (**3a**–**e**) and 1,4-disubstituted triazole (**4a**–**e**) were synthesized and characterized as described previously by Aissa et al. [21]. Imatinib mesylate (STI571), used as a reference therapeutic molecule, Ficoll–Hypaque and 3-(4,5-dimethylthiazol-2-yl)-2,5-diphenyl-2*H*-tetrazolium bromide (MTT) were purchased from Sigma-Aldrich (St. Louis, MO, USA).

### 2.2. Cell Lines

The K562 cell line was used as a cellular model of CML for screening. Cells (kindly provided by José Luis, Université d’Aix-Marseille, Marseille, France) were cultured in complete RPMI (Roswell Park Memorial Institute) 1640 medium, supplemented with 10% fetal bovine serum, 1% glutamine and 1% penicillin/streptomycin under an atmosphere of 5% CO_2_ and 95% air at 37 °C.

### 2.3. Patient Leukemia Cells

Primary CML cells were obtained from the peripheral blood of 15 CML patients including 10 patients with untreated chronic phase and 5 patients with resistance against imatinib or multiple TKIs. The CML patients were divided into 2 groups as follows:**Group 1**: Contained 10 CML patients at diagnosis;**Group 2**: Contained 5 CML patients resistant to imatinib or multiple TKIs with a bcr–abl/abl ratio > 10% at 3 months or >1% at 6 months.

All patients’ samples were collected during routine examinations at the Hematology Department at Pasteur Institute of Tunis, Tunisia. Written informed consent was obtained from each patient for blood collection and use of their clinical data and cells.

The clinical characteristics of patients are shown in Table 1.

### 2.4. Peripheral Blood Mononuclear Cells Isolation

Whole blood from patients was collected into vacuum tubes containing anticoagulant, and peripheral blood mononuclear cells (PBMCs) were isolated by the Ficoll–Hypaque density gradient centrifugation method using a Histopaque-1077 (Sigma Diagnostics, Inc., Budapest, Hungary). Cells at the interface were collected, washed twice and resuspended in complete RPMI medium. The trypan blue dye exclusion test was used to determine cell viability. All experiments were performed when cell viability was above 95%.

### 2.5. Antitumor Assay

#### 2.5.1. Cytotoxicity Assay

K562 cell viability was determined using the MTT assay [22]. Cells were seeded in 96-well plates at an appropriate density (i.e., 6000 cells per well) and were allowed to settle for 24 h before treatment. Next, cells were treated with the newly synthesized derivatives, **3a**–**e** and **4a**–**e**, at different concentrations (0–200 µg/mL) for 24 h. After incubation, 20 µL of 3-(4,5-dimethyl-2-thiazolyl)-2,5-diphenyl-2*H*-tetrazolium bromide solution (MTT) of 5 mg/mL concentration was added to each well. After 4 h of incubation at 37 °C, the plates were centrifuged (1200 rpm, 3 min, room temperature). The supernatants were discarded, and then 200 μL of dimethyl sulfoxide (DMSO) were added to dissolve the formazan crystals. The optical density (OD) was measured using the Multiskan Microplate Spectrophotometer (Thermo Fisher Scientific, Waltham, MA, USA) at a wavelength of 560 nm. The results are expressed as the percentage of viable cells relative to the negative control (untreated cells) according to the following formula: OD (experiment samples) − OD (blank)/[OD (control) − OD (blank)] × 100%. The IC_50_ values were determined as the concentration that reduced the cell viability by 50% for each sample. Two independent experiments were performed in triplicate.

#### 2.5.2. Anti-Proliferative Assay

The anti-proliferative activity was evaluated on K562 cells and CML primary cells using the MTT assay as described previously [22]. The K562 cells (3000 cells per well) were allowed to settle overnight before being treated by the synthesized compounds **3a**–**e** and **4a**–**e** for 72 h at different concentrations (0–100 µg/mL). Based on the anti-proliferative activity of the different derivatives, on K562 cells, the most potent compounds were chosen to evaluate their effect on PBMCs isolated from CML patients.

The CML primary cells (5 × 10^5^ cells per well) were treated with imatinib mesylate (IM) at 10 µM (positive control) and/or with the selected synthesized compounds at different concentrations (0–100 µg/mL). The proliferation in each sample is expressed as a percent with respect to the untreated cells (negative control). The concentration required for 50% inhibition of the cellular proliferation (IC_50_) was calculated for each sample. Two independent experiments were performed in triplicate.

### 2.6. Apoptosis-Induced Detection

To determine the apoptosis induction activity in 3d-treated K562 cells, the RealTime-Glo™ Annexin V Apoptosis and Necrosis Assay (Promega) was used in accordance with instructions from the manufacturer. Briefly, cells were seeded in 50 µL growth medium in white 96-well microtiter plates (10^4^ cells/ well). A set of control wells without cells present (growth medium only) to determine background luminescence and background fluorescence was included. The appropriate dilution of the compound **3d** was prepared in the growth medium at 4-fold the desired final concentration. A set of control wells (untreated controls) was included, and staurosporine (1 µM) was used as an apoptosis inducer (positive control). Cells were incubated for 24 h at 37 °C and 5% CO_2_ in a humidified cell culture incubator. Luminescence and fluorescence (Ex 475, Em 500–550) were measured.

### 2.7. Intracellular ROS Detection

The total intracellular superoxide and hydroxyl radicals in the K562 cells was detected using the Cellular Reactive Oxygen Species Detection Assay Kit (ab186027, Abcam, Cambridge, UK), following the manufacturer’s instructions. Briefly, the K562 cells were plated (5 × 10^4^/100 µL per well) in a 96-well plate overnight. Subsequently, cells were treated with different doses of compound **3d** and stained with ROS red stock solution for 2 h at room temperature. After incubation, a fluorescence microplate reader (PerkinElmer, Waltham, MA, USA) was used to detect the intensity of the fluorescence at 520 nm and 605 nm excitation and emission wavelengths, respectively.

### 2.8. Western Blot Analysis

The K562 cells were treated with compound **3d** at different concentrations (i.e., 22.5, 45 and 90 µM). After incubation, cells were lysed in Laemmli buffer after being washed with PBS twice. After separation and transfer using 12% Tris-glycine SDS-polyacrylamide gels and PVDF membranes, samples were blocked using 5% milk for 2 h. Then, PVDF membranes were incubated overnight at 4 °C, with the following primary Rabbit monoclonal antibodies: Akt, Phospho-Akt (Ser473), p44/42 MAPK (Erk1/2), Phospho-p44/42 MAPK (Erk1/2) (Thr202/Tyr204), p38 MAPK and Phospho-p38 MAPK (Cell Signaling). GAPDH antibody was used in a 1:1000 dilution for equal loading control.

After incubation with secondary antibody (Promega) for 1 h at room temperature, membranes were visualized using the enhanced chemiluminescence kit (Pierce, IL, USA). ImageJ (Media Cybernetics, MD, USA) was used for the quantitative analysis.

### 2.9. Statistical Data Analysis

Statistical analyses were performed using the GraphPad Prism6 scientific software (GraphPad, Software, Inc., La Jolla, CA, USA). Data are expressed as the mean ± SD of a minimum of two independent experiments. One-way ANOVA with a Tukey post hoc test was used to compare the differences between the experimental groups. The *p*-values of <0.05 (*), <0.01 (**) and <0.001 (***) were considered to be statistically significant.

All experiments were repeated at least two times.

## 3. Results and Discussion

### 3.1. Cytotoxic Effect on K562 Cells

The effect of different concentrations of tyrosol (**1**), tyrosol-alkyne (**2**) and the isoxazole (**3a**–**e**) and triazole (**4a**–**e**) derivatives (Table 2) on K562 cell viability was evaluated using the MTT assay.

The results showed that after 24 h of treatment, all the synthesized compounds decreased the viability of the K562 cells in dose-dependent manner (Figure 1).

Notably, tyrosol-alkyne (**2**) as well as synthesized cycloadducts (isoxazoles **3a**–**e** and triazoles **4a**–**e**) exhibited greater potent cytotoxic effects than the starting compound 1. This finding showed that the propargylation of the phenolic function, as well as the junction of isoxazole or triazole fragments to compound **1**, considerably improved the cytotoxic effect against the K562 cells. Particularly, it is interesting to note that the tyrosol-coupled 3,5-disubstituted isoxazole derivatives (**3a**–**e**) (IC_50_ values ranging from 74 µg/mL to 190 µg/mL) showed a greater cytotoxic effect against the K562 cells than the tyrosol-coupled 1,4-disubstituted triazole ones (**4a**–**e**) (IC_50_ values greater than 200 μg/mL).

### 3.2. Anti-Proliferative Effect on K562 Cells

A dose-dependent inhibition was observed for all derivatives after 72 h of treatment (Figure 2). Our results are in accordance with previous studies, which reported the anti-proliferative activity of triazole and isoxazole derivatives against various human leukemia cell lines such as K562, MV4-11 and HL60 [15,16,17,18].

Interestingly, we noticed that the newly synthesized derivatives with a junction of tyrosol **1** to isoxazole (**3a**–**e**) or triazole (**4a**–**e**) improved their anti-proliferative activity at lower concentrations.

Moreover, we found that tyrosol-coupled 3,5-disubstituted isoxazole derivatives (**3a**–**e**) (IC_50_ values ranging from 16 to 24 µg/mL) exhibited a higher anti-proliferative effect against K562 cells than tyrosol-coupled 1,4-disubstituted triazole (**4a**–**e**) (IC_50_ values ranging from 18 to 50 μg/mL) (Table 3).

Especially, the 3,5-disubstituted isoxazole compounds **3a** (4-OCH_3-_C_6_H_4_), **3d** (4-*t*-Bu-C_6_H_4_) and **3e** (4-Cl-C_6_H_4_) displayed the highest anti-proliferative activity towards the K562 tumor cells with IC_50_ values of 18 (55 µM), 16 (45 µM) and 18 µg/mL (54.5 µM), respectively. These results show the effect of the substituent introduced into these derivatives, **3a**, **3d** and **3e**, in improving the anti-proliferative activity compared to the unsubstituted analog **3b** (IC_50_ = 81 µM). The *t*-Bu fragment in compound **3d** exerting a significant positive inductive effect (+I), seemed to be at the origin of higher activity compared to the 4-OCH_3_ and 4-Cl substituents in compounds **3a** and **3e**, which exerted mesomeric donor (+M) and negative inductive (−I) effects, respectively. The size of these substituents and their effect on the free rotation of the aromatic nucleus to which they were attached as well as on the conformation of the entire molecule could also partly explain this difference in activity.

In the series of the 1,4-disubstituted triazoles derivatives (**4a**–**e**), compounds **8a** (-C_6_H_5_) and **4e** (2,4,5-trichlo-C_6_H_2_) exhibited the most potent anti-proliferative activity against the K562 tumor cells with IC_50_ values of 18 (61 µM) and 25 µg/mL (63 µM), respectively. Thus, the nonsubstitution (**4a**) or the polychlorination (**4e**) of the aromatic ring carried by the triazole moiety are clearly in favor of the anti-proliferative activity. The free rotation around the bond connecting the unsubstituted aromatic ring to the triazole in **4a** could promote a good interaction between this whole moiety and target proteins in K562 cells. On the other hand, the trichlorination of the aromatic ring attached to triazole in **8e** and the negative inductive (−I) and donor mesomeric (+M) effects exerted by the three chlorine atoms in specific positions (2, 4, 5) were certainly behind its potent activity compared to that of its monochlorinated analog **4b** (4-Cl) (IC_50_ = 72.5 µM).

In agreement with our previous study, compounds with methyl, methoxy or chlorine substitution at R groups in isoxazole derivatives and compounds with a chlorine substitution at R groups in triazole derivatives were more effective against glioblastoma cells [21].

Based on the promising antitumor effect against K562 cells, compounds **3d** and **4a** were chosen for deeper evaluation of their anti-proliferative activity on PBMCs isolated from CML patients.

### 3.3. Effect on Leukemic PBMCs 

Commercial cell lines have been engineered to continuously and consistently divide; therefore, they do not fully represent primary leukemia cell characteristics. For this reason, it was necessary to confirm the anti-proliferative effect of the most potent compounds, **3d** and **4a**, against K562 cells on ex vivo patient samples.

Primarily, patients were classified into two groups: the first group represented patients affected by CML at diagnosis, with a median age of 50 years. Their mean white blood cell count (WBC) was 156,413/mm^3^ with a range from 40,000 to 647,380/mm^3^; the second group represented CML patients with resistance against imatinib or multiple TKIs. For this group, the median age was 55 years and patients’ BCR–ABL ratio ranged from 20% to 64% at 3 months and from 7% to 32% at 6 months.

Primary CML cells of all patients were treated with imatinib mesylate (IM) at 10 µM (positive control) and/or one of the selected compounds (i.e., **3d** and **4a**) for 72 h. Our data revealed a large variability in the proliferation inhibition among all patients after treatment with IM (10 µM) (Table 4).

In the first group, the data show that the percentage of inhibition ranged from 15.3% to 50.9% with only IM (10 µM) treatment (Table 4). The CML cells treated with the compound **3d** alone decreased cell proliferation in a concentration-dependent manner with IC_50_ values ranging from 23 to 283 µM (Figure 3). It is worth noting that the potential anti-proliferative activity of the derivative **3d** did not correlate with the different BCR–ABL transcripts, since the majority of CML patients in this group represented b3a2 and/or a b2a2 junction, except one patient with a e1a2 BCR–ABL fusion.

Compared to **3d**, the compound **4a** represented a lower effect on CML patients’ cells. Indeed, **4a** showed slight anti-proliferative activity even at high concentrations (340 µM), except one patient, **P7**, with an IC_50_ value of 76 µM (Figure 3).

On the other hand, we examined the combined effect of IM at 10 µM with different concentrations of **3d** or **4a** on all patients’ PBMC proliferation for 72 h. We observed that, only for three patients (**P4**, **P7**, and **P10**), the co-treatment with IM with the selected compounds significantly potentiates the anti-proliferative effect (Figure 4).

In the second group, the treatment with IM (10 µM) only caused a decrease in the cell proliferation from 21.3% to 35.9% (Table 4). Interestingly, a dose-dependent anti-proliferative activity was obtained in presence of the compound **3d** with the IC_50_ values ranging from 70 µM to 184.4 µM, whereas the compound **4a** represented a low anti-proliferative activity (38% of inhibition) even at high concentrations (Figure 3).

The potential anti-proliferative activity of the compound **3d** did not depend on the patients’ BCR–ABL ratio. In fact, two patients with the same BCR–ABL ratio presented different IC_50_ values.

It is worth noting that all the patients in this group represented the most common BCR–ABL transcripts of CML (i.e., b2a2 and (b3a2) in which the compound **3d** exerted a dose-dependent inhibition effect, except for in the patient presenting a T315I mutation (IC_50_ > 300 µM). Our results are consistent with those of Zhang et al., who reported that the mutation of threonine by an isoleucine in position 315 (T315I) can lead to resistance to most TKIs, and it is only sensitive to ponatinib [23]. In addition, it was reported that heterocyclic compounds inhibited CML cells, including those expressing the CML T315I mutation, at nanomolar concentrations [24]. These results pave the way for further development of this class of compounds.

On the other hand, we noticed that the synthetic compounds did not potentialize the effect of the IM (10 µM) on the PBMCs of this patient group. Further studies are necessary to assess the effect of different combinations on a higher number of resistant patients.

### 3.4. Compound **3d** Induced Apoptosis in K562 Leukemia Cells

Based on its interesting anti-proliferative effect against K562 and primary CML cells, the compound **3d** was selected for deeper investigation in order to identify its mechanism of action. We found that exposure of the K562 cell line to the **3d** compound caused a significant increase in the luminescence signal in a dose-dependent manner, compared with untreated control (Figure 5). This result indicated that **3d** compound induced apoptosis in the K562 cells. This finding is in agreement with our previous study, showing that the molecule triggers anti-proliferative effects in U87 cells through the induction of apoptosis [21]. Our results are also consistent with other studies, which reported that isoxazole derivatives act as an anticancer agent inducing apoptosis [15,16].

### 3.5. Compound **3d** Triggered ROS Production in K562 Leukemia Cells

Many ROS-modulating agents have been developed to induce oxidative stress in anticancer therapy by accessing the ROS tolerance in cancer cells. The rationale is that drug-induced excessive oxidative stress frequently triggers DNA damage and apoptosis [25]. To investigate whether oxidative stress played a pivotal role in **3d** anti-proliferative activity, we performed an ROS assay. After **3d** treatment, we observed an increase in ROS production in a dose-dependent manner (Figure 6). These results suggest that ROS markedly contributed to the anticancer effect of **3d** in the K562 cancer cells. A previous study showed that two isoxazole compounds (i.e., **5m** and **5o**) caused the generation of ROS, induction of apoptotic cell death and cell cycle arrest at different phases in liver cancer cells [26]. In accordance with previous studies, our findings indicate that the **3d** compound induced apoptosis via an ROS-mediated signaling pathway.

### 3.6. Effect of 3d Treatment on MAPK, Erk and Akt Activation

ROS appears as one of the main mechanisms by which most anticancer drugs exert their antineoplastic effects, particularly through the activation of MAPKs and leading to cell growth arrest. Based on the fact that the **3d** compound affected K562 cell proliferation and induced apoptosis by promoting ROS accumulation, the MAPK protein family members were analyzed. Compared with untreated K562 cells, the **3d**-treated cells revealed statistically significant upregulation of Erk 1/2 (extracellular signal-regulated kinases), Akt and p38-MAPKs (Figure 7) phosphorylation. Thus, we can infer that **3d** induces ROS generation, which then affects K562 cancer cell proliferation by modulating signaling transduction, specifically by ERK, p-38 and Akt dephosphorylation.

## 4. Conclusions

In this study, we demonstrated the inhibitory effect of tyrosol derivatives bearing 3,5-disubstituted isoxazole and 1,4-disubstituted triazole on both K562 and primary chronic myeloid leukemia cell proliferation. The results revealed that the isoxazole derivative 2-(4-((3-(4-(tert-butyl)phenyl)isoxazol-5-yl)methoxy)phenyl) ethanol (**3d**) was the most active compound among these novel tyrosol derivatives. Overall, the results from our analyses showed that the total blockade of Akt and ERK1/2 phosphorylation contributed to its pro-apoptotic effect.

This compound has previously been reported specifically for glioblastoma cancer cells, but the new data suggest a broader use for this analogue as a promising lead for further developments in the fight against solid tumors as well as hematological malignancies including those that have become resistant to current treatments (TKI). Further investigations are necessary to set-up the direction for the design and development of this compound as a single agent or in combination with a targeted therapy.

## Figures and Tables

**Figure 1 molecules-27-05086-f001:**
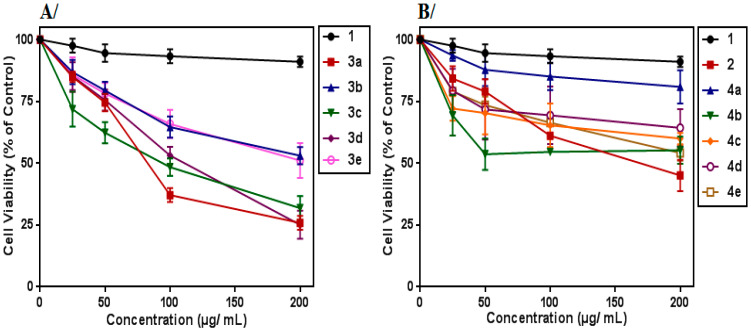
Tyrosol derivatives affected K562 cells’ viability: (**A**) K562 cells treated with tyrosol (**1**) and the synthesized isoxazoles (**3a**–**e**); (**B**) cells treated with tyrosol (**1**), tyrosol-alkyne (**2**) and the synthesized triazoles (**4a**–**e**). Cells were treated with increasing concentrations for 24 h, and their viability was determined by the metabolic rate using the MTT assay. Absorbance values were measured at 560 nm and normalized against untreated cells. Data represent the mean ± SEM of three independent experiments performed in triplicate. All data were statistically significant (*p* < 0.05), except tyrosol (1) (at 25, 50 and 100 µg/mL) and **4a** at 25 µg/mL.

**Figure 2 molecules-27-05086-f002:**
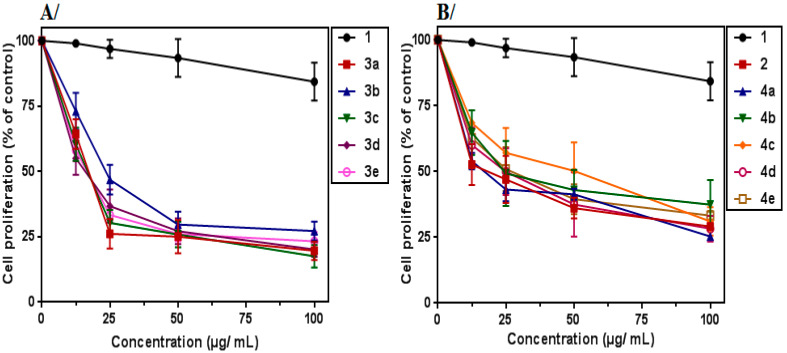
Effect of tyrosol (**1**) and its derivatives on K562 cell proliferation: (**A**) K562 cells treated with tyrosol (**1**) and the synthesized isoxazoles (**3a**–**e**) for 72 h; (**B**) K562 cells treated with tyrosol (**1**), tyrosol-alkyne (**2)** and the synthesized triazoles (**4a**–**e**) at various concentrations for 72 h. Data represent the mean ± SEM of three independent experiments performed in triplicate. All data were statistically significant (*p* < 0.05), except tyrosol (**1**) at 12.5, 25 and 50 µg/mL.

**Figure 3 molecules-27-05086-f003:**
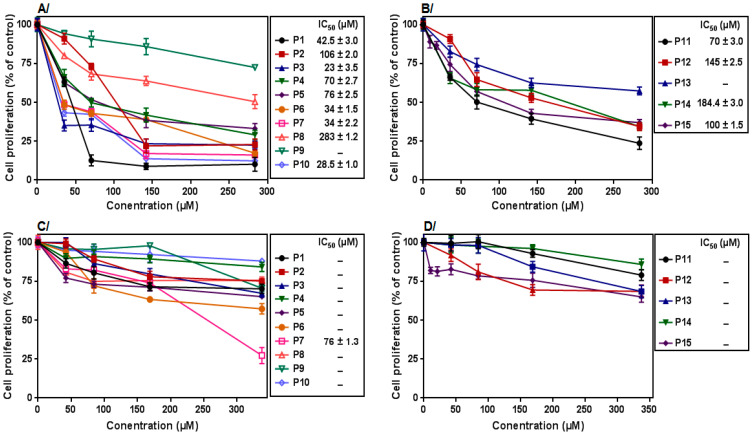
Anti-proliferative activity of compounds **3d** and **4a** on primary CML cells: (**A**) cells isolated from patients at diagnosis (group 1) treated with compound **3d** for 72 h; (**B**) cells isolated from resistant patients (group 2) treated with compound **3d**; (**C**) cells isolated from group 1 treated with compound **4a**; (**D**) cells isolated from group 2 treated with compound **4a**. Data represent the mean ± SEM of three independent experiments performed in triplicate. All data were statistically significant (*p* < 0.05).

**Figure 4 molecules-27-05086-f004:**
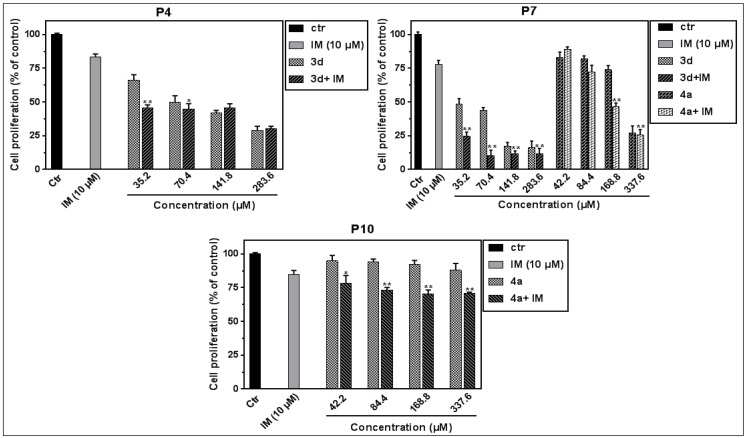
Effect of the compound **3d**, **4a** and their combination with IM (10 µM) on patients’ PBMC proliferation. Comparison between the percentage of PBMC proliferation in PBMC treated only with **3d** or **4a** and cotreated with IM (10 µM). * *p* < 0.05 and ** *p* < 0.01 values; used to compare the percentage of cell proliferation treated with **3d** or **4a** alone or combined with IM (10 µM). Data represent the mean ± SEM of two independent experiments performed in triplicate.

**Figure 5 molecules-27-05086-f005:**
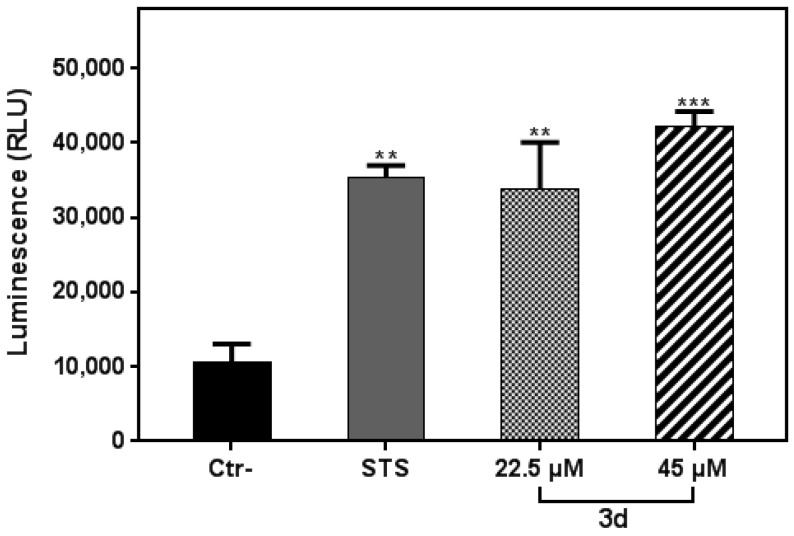
Apoptosis induction in K562 cells treated with **3d** for 24 h in the presence of RealTime-GloTM Annexin V Apoptosis and Necrosis Assay reagent. RLU (left panels, PS: annexin V binding) was collected. Staurosporine (STS) was used as the positive control. Data represent the mean of 2 readings for each replicate ± SD. All data were statistically significant (*p* < 0.05). **, and *** denote *p* < 0.01, and 0.001, respectively.

**Figure 6 molecules-27-05086-f006:**
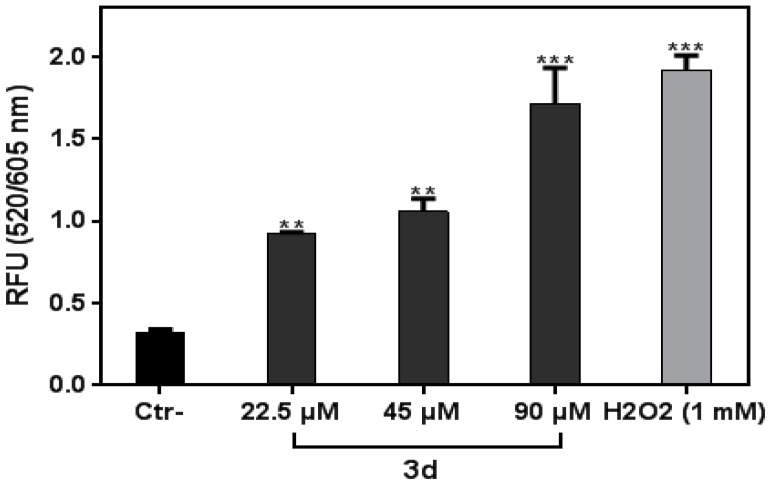
The compound **3d** induced reactive oxygen species production in K562 cells. Cells were treated with **3d** at different concentrations for 2 h. H_2_O_2_ (1 mM) was used as the positive control. Data are the mean of two independent experiments. All data were statistically significant (*p* < 0.05). **, and *** denote *p* < 0.01, and 0.001, respectively.

**Figure 7 molecules-27-05086-f007:**
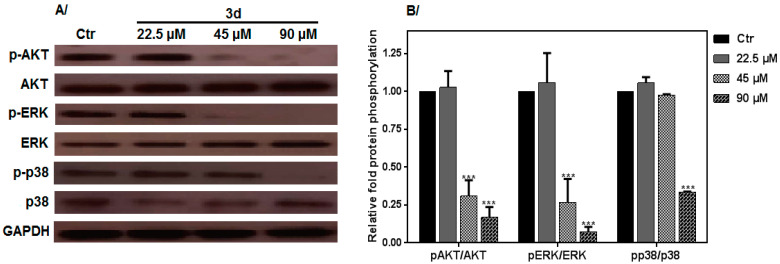
Western blot analysis: (**A**) K562 cells were treated with compound **3d** for 1 h to evaluate the MAPK phosphorylation AKT, ERK1/2 and p-38; (**B**) relative protein phosphorylation/expression level. GAPDH was used as the internal standard; *p* < 0.05 was used to compare to the control group. All data were statistically significant (*p* < 0.05) compared to the control (untreated cells). *** denote *p* < 0.001.

**Table 1 molecules-27-05086-t001:** Clinical features of the CML patients.

N	Age	Sex	Diagnosis	Phase	Variant BCR–ABL	BCR–ABL/ABL Ratio	White Cell Count/mm^3^	Platelets/mm^3^
**Group 1**								
**1**	60	M	CML	At diagnosis	b2a2		154,620	_
**2**	24	M	CML	At diagnosis	b2a2		157,000	98,000
**3**	51	M	CML	At diagnosis	b3a2		95,590	629,000,000
**4**	49	M	CML	At diagnosis	b2a2		647,380	407,000
**5**	61	F	CML	At diagnosis	b2a2/b3a2		40,000	281,000
**6**	_	F	CML	At diagnosis	e1a2 (P190)		50,000	219,000
**7**	42	F	CML	At diagnosis	b2a2		127,000	436,000
**8**	47	F	CML	At diagnosis	b3a2		60,240	454,000
**9**	54	M	CML	At diagnosis	b2a2 (T315I?)		132,300	384,000
**10**	60	F	CML	At diagnosis	b2a2		100,000	120,000
**Group 2**								
**11**	48	F	CML	Resistant	b2a2	3 months 41% 6 months 32%	4630	196,000
**12**	61	F	CML	Resistant	b2a2	64%	71,520	82,000
**13**	55	F	CML	Resistant	b3a2			
**14**	52	M	CML	Resistant	b2a2	3 months 22% 6 months 7%	218,000	_
**15**	60	F	CML	Resistant	b2a2	3 months 20% 6 months 7%	4600	80,000

CML: chronic myeloid leukemia.

**Table 2 molecules-27-05086-t002:** Structures of tyrosol (**1**), tyrosol-alkyne (**2**), tyrosol-linked 3,5-disubstituted isoxazoles (**3a**–**e**) and tyrosol-linked 1,4-disubstituted triazoles (**4a**–**e**).

Compound	Chemical Formula
**1**	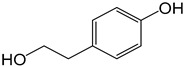
**2**	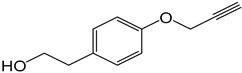
**3a**	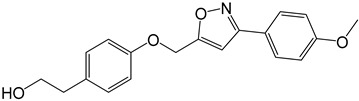
**3b**	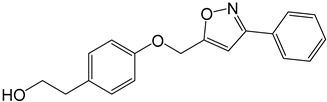
**3c**	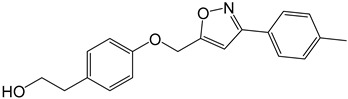
**3d**	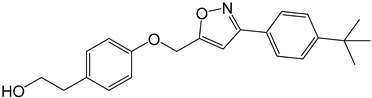
**3e**	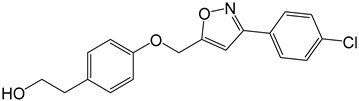
**4a**	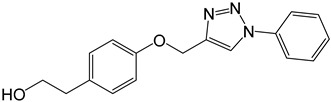
**4b**	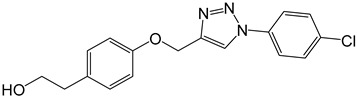
**4c**	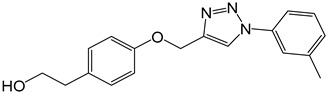
**4d**	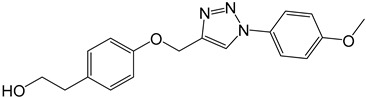
**4e**	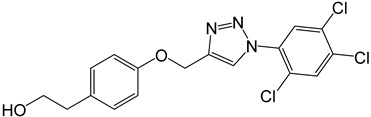

**Table 3 molecules-27-05086-t003:** Anti-proliferative activity of compounds tyrosol-alkyne (**1**), cycloadducts (**2**), (**3a**–**e**) and (**4a**–**e**) on the K562 tumor cell line.

Compound	1	2	3a	3b	3c	3d	3e	4a	4b	4c	4d	4e
**IC_50_ (µg/mL)**	>100	18 ± 1.5	18 ± 2.0	24 ± 0.9	18 ± 3.0	16 ± 1.0	18 ± 2.5	18 ± 0.8	24 ± 2.0	50 ± 1.5	25 ± 1.2	25 ± 0.5
**IC_50_ (µM)**	>400	101 ± 8.4	55 ± 6.1	81 ± 3	58 ± 9.6	45 ± 2.8	54.5 ± 7.5	61 ± 2.7	72.5 ± 6	161 ± 4.8	76.5 ± 3.6	63 ± 1.2

**Table 4 molecules-27-05086-t004:** Inhibition of primary CML cell proliferation in the presence of IM (10 µM).

Patient	% Inhibition
** Group 1 **	
**1**	34.8 ± 0.84
**2**	27.3 ± 2.2
**3**	23.8 ± 1.0
**4**	16.4 ± 0.8
**5**	29.9 ± 0.5
**6**	50.9 ± 1.3
**7**	22.1 ± 0.9
**8**	23.4 ± 1.2
**9**	21 ± 2.4
**10**	15.3 ± 3.3
** Group 2 **	
**11**	35.9 ± 1.5
**12**	21.3 ± 2.1
**13**	22.1 ± 0.8
**14**	21.8 ± 2.9
**15**	30.6 ± 0.6

## Data Availability

The data presented in this study are available in this article.

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
