# Peer review of "Tyrosol Derivatives, Bearing 3,5-Disubstituted Isoxazole and 1,4-Disubstituted Triazole, as Potential Antileukemia Agents by Promoting Apoptosis"

_molecules, 2022, doi:10.3390/molecules27165086_

Round 1
Reviewer 1 Report
The submitted manuscript is revised well and its publishible to this journal.
Author Response
Authors thank the reviewer for the positive comments, which have helped us to improve the manuscript significantly.
Reviewer 2 Report
The authors performed additional experiments and therefore implemented their manuscript with new information about their compounds. Most of the concerns have been addressed, although the chemical structures still seem to be stretched (but it might also be a problem in the format after uploading the files).
I would now recommend publication of the manuscript in its current form
Author Response
Authors thank the reviewer for the positive comments.
Point 1. Most of the concerns have been addressed, although the chemical structures still seem to be stretched (but it might also be a problem in the format after uploading the files).
Response 1. The chemical structures are now improved in the revised manuscript.
Reviewer 3 Report
The article has been written well. The subjects of triazoles and isoxazoles are interesting. I strongly recommend the publication of this article after a minor revision.
1. Please redraw the chemical structure of compounds in the table 2.
2. The article is about 3,5-disubstituted isoxazole and 1,4- disubstituted triazole! But there is no information about these structures in the introduction! Please add a new paragraph about this subject by adding relevant review articles and articles. Please check https://onlinelibrary.wiley.com/doi/abs/10.1002/aoc.4446 to find the importance of 1,4- disubstituted triazoles in the synthesizing diverse range of hybrid molecules. I also suggest to the authors to check this review article https://www.ingentaconnect.com/content/ben/mrmc/2021/00000021/00000005/art00003 focusing on importance of triazoles in synthesizing new types of molecules. If the names of these compounds are in the title of article and if the authors think these compounds are important they should add a paragraph by inserting relevant articles and review articles.
3. There is no need for some sentences in the abstract. Authors can remove them such as "Chronic myeloid leukemia (CML) is a myeloproliferative disorder, characterized by the presence of the Bcr-Abl fusion oncogene." The abstract should be short with important findings of a research article.
Author Response
"Please see the attachment."

This manuscript is a resubmission of an earlier submission. The following is a list of the peer review reports and author responses from that submission.
Round 1
Reviewer 1 Report
The submitted manuscript is well written and publisible after the minor suggestions. Authors need to modify their manuscript as according to belows points.
- Introduction need to improve much because the objective of the work and behind the story is not clear.
- Table 1. need to add at the end.
- if possible authors need to make a graph plot (sigmoidal curve) of IC50 values for table table 4 and 5 for group 1 and 2. Also construct a linear plot for these values. it will be better for the readers. they can understand well.
- Conclusion need to elaborate more with the experimental observations. also add new heading for the conclusion.
Reviewer 2 Report
The research work of Abdelkafi-Koubaa et al reports the biological characterization of some isoxazole- and 1,2,3-triazole-containing compounds, endowed with antiproliferative activity and also active against human patient leukemia cells. The paper is well written and the conclusions are supported by the data.
The authors made it clear that the compounds have already been synthesized in a different paper work. Nevertheless, the numbers of the compounds should not refer to the previous work, so compounds 6a-e and 8a-e should be labelled as 3a-e and 4a-e. The structures of the compounds have some wrong bond angles. Please, clean up structures and format them all with the same style. Although specified, the tables reporting IC50 and activities should be presented with the standard deviations, since all the test have been performed at least in triplicates.
In general, I would also made additional comments about the fact that these compounds were reported as specific for glioblastioma cancer cells but the new data suggest a broader use for these analogues, as good starting point for further developments in the fight against cancer and leukemia.
Minor checks - line 172: put “their” in place of “its”; line 200: 8e (2,4,5-trichlo-C6H4) should be 8e (2,4,5-trichlo-C6H2); line 233: “the percentage of inhibition of is ranging” should be “the percentage of inhibition is ranging”.
Therefore, I would recommend publication in the journal after addressing the abovementioned issues.
Reviewer 3 Report
The authors focused only on Human Leukemia Cells experiments, I recommend by adding synthetic data and other experiments to submit to other suitable journals. On the basis of lack of scientific data, I must reject this manuscript.
Round 2
Reviewer 3 Report
The authors do not add experimental synthetic data, only a few changes in the text...